A divergence between underlying and final causes of death in selected conditions: an analysis of death registries in Peru

Carrillo-Larco Rodrigo M. 1 2 r.carrillo-larco@imperial.ac.uk
http://orcid.org/0000-0002-6834-1376 Bernabe-Ortiz Antonio 2 3 4 antonio.bernabe@upch.pe
1 Department of Epidemiology and Biostatistics, Imperial College London , London , UK
2 CRONICAS Center of Excellence in Chronic Diseases, Universidad Peruana Cayetano Heredia , Lima , Peru
3 Faculty of Epidemiology and Population Health, London School of Hygiene & Tropical Medicine, University of London , London , UK
4 Facultad de Ciencias de la Salud, Universidad Peruana de Ciencias Aplicadas , Lima , Peru
Cooper Richard
Electronic publication date: 2018 Nov 12
Publication date: 2018
Volume: 6
Electronic Location ID: e5948
Received 2018 Aug 14; Accepted 2018 Oct 17
Copyright: © 2018 Carrillo-Larco and Bernabe-Ortiz
Copyright year: 2018
Copyright holder: Carrillo-Larco and Bernabe-Ortiz
License: This is an open access article distributed under the terms of the Creative Commons Attribution License, which permits unrestricted use, distribution, reproduction and adaptation in any medium and for any purpose provided that it is properly attributed. For attribution, the original author(s), title, publication source (PeerJ) and either DOI or URL of the article must be cited.
License URL: https://creativecommons.org/licenses/by/4.0/

Keywords: Cause of death, Mortality, Cardiovascular diseases, Non-communicable diseases, Communicable diseases, Peru

Funding: Wellcome Trust (103994/Z/14/Z) Antonio Bernabe-Ortiz is a Research Training Fellow in Public Health and Tropical Medicine funded by the Wellcome Trust (103994/Z/14/Z). The funders had no role in study design, data collection and analysis, decision to publish, or preparation of the manuscript.

==============================
Background

The underlying cause of death is used to study country and global mortality trends and profiles. The final cause of death could also inform the ultimately cause of death in individuals with underlying conditions. Whether there is a pattern between the underlying and final cause of death has not been explored using national death registries. We studied what final causes of death were most common among selected underlying causes using national death registries in Peru, 2015.

Methods

Underlying and final causes of death were classified according to their ICD-10 codes. Underlying causes included chronic kidney disease (CKD), chronic obstructive pulmonary disease (COPD), hypertension (HTN), diabetes, and selected cancers (cervix, breast, stomach, prostate, and lung). Final causes were categorized as: communicable, cardiovascular, and cancers. Descriptive statistics were used.

Results

A total of 77,065 death registries were analyzed; cases had a mean age of 69.4 (SD: 19.3) years at death and were mostly men (53.9%). When the underlying cause was HTN, the most frequent final cause was cardiovascular diseases (82.3%). For all the other underlying causes, the most frequent final cause was communicable diseases: COPD (86.4%), CKD (79.3%), cancer (76.5%), and diabetes (68.3%).

Conclusions

In four selected underlying causes of death there was a divergence with respect to the final cause, suggesting there was a shift from non-communicable to communicable causes. Although efforts should be deployed to prevent underlying non-communicable diseases, potential communicable complications should not be neglected.

Introduction

Global reports use the underlying cause of death to inform death trends and profiles (GBD 2016 Causes of Death Collaborators, 2017). This approach is relevant from a public health perspective because delaying or avoiding the underlying cause would prevent the chain of events leading to death. However, from a public health and clinical perspective, understanding the final cause of death is also relevant. For example, whether a patient with type 2 diabetes is likely to finally perish of a coronary event or sepsis could guide what information should be provided to these patients (i.e., patient empowerment) and to health personnel looking after them. Information could thus target prevention of non-communicable diseases, communicable disease, or both. From a clinical perspective, understanding the patterns between underlying and final cause of death could provide clinicians with evidence on what to prevent during hospitalizations or emergency visits. Finally, understanding the underlying-final cause of death profile could provide additional insights into the epidemiological transition. Although most countries now face a large (underlying) mortality burden of non-communicable diseases (GBD 2016 Causes of Death Collaborators, 2017; GBD 2016 DALYs and HALE Collaborators, 2017; GBD 2016 Disease and Injury Incidence and Prevalence Collaborators, 2017), hence allocating resources to prevent them, understating what the final causes of death are could help to prevent complications, that is, final causes of death. To the best of our knowledge, although multiple countries and international groups have made relevant efforts to improve and report underlying causes of death, little has been done to study the final causes of death and how these interact with the underlying ones. Consequently, we aimed to describe the final causes of death of selected underlying causes using national death registries in Peru, 2015.

Methods

Data sources and subjects

Death registries were requested from the Peruvian Ministry of Health (Ministerio de Salud del Perú, 2018a). These records compile death certificates from all health systems in Peru, though underreporting could be expected from facilities outside the ministry system (e.g., private clinics). Available data included underlying and final cause of death (ICD-10 codes), geographic location, sex, and age at death.

For this study people aged 18 years or above were included, because most of the diseases of interest (e.g., type-2 diabetes mellitus) are uncommon in children. In addition, records in which the final cause was cardiac arrest (ICD-10: I48), chronic kidney disease (CKD) (ICD-10: N18), type 2 diabetes mellitus (ICD-10: E10, E11, E13), chronic obstructive pulmonary disease (COPD) (ICD-10: J41, J42, J44), hypertension (HTN) (ICD-10: I10), and selected cancers (ICD-10: C16 (stomach), C61 (prostate), C34 (bronchus and lung), C53 (cervix uteri), C50 (breast)), were excluded. Cardiac arrest cases in the final cause of death were excluded to improve the data quality, as it is not a clearly defined diagnosis. The other conditions, which were also the underlying causes of interest, were excluded to improve data quality and avoid overlapping between underlying and final cause of death, that is, a registry with the same underlying and final cause which could imply data quality issues or an unintended error when filling in the certificate. The number of observations excluded at each step of data cleaning is depicted in Fig. S1.

Characteristics of data sources

Although the analyzed death registries have a national coverage, there could have been relevant under-reporting. In 2011 the coverage of the death registry was 53.8%; in other words, roughly half of all the expected deaths were registered (Ministerio de Salud del Perú, 2018b). Moreover, there was wide variation across regions, ranging from 21.9% to 80.7% coverage (Ministerio de Salud del Perú, 2018b). Fortunately, though to the best of our knowledge it has not been quantified and reported, these coverage estimates have improved. At some extent, this argument has been supported by coverage reported by the Global Burden of Disease team (GBD 2016 Causes of Death Collaborators, 2017).

This underreporting would have implications on the results, interpretation and impact of this work. First, the findings should be understood in the context of the underreporting issue, meaning these findings would not tell a complete picture but rather an introduction of what could be happening. Nonetheless, the results could tell a story that needs further verification, fostering the improvement of available registries as it has been done with the introduction of online registries in Pardo et al. (2015). Second, and with regards to this work and the communicable diseases outcome, these estimates could be under-reported assuming worse coverage in resource-limited settings, which would be where most communicable (infectious) diseases occur. If this is the case, we could expect higher rates of communicable disease deaths.

Variables

We studied two causes of death as reported on death certificates based on ICD-10 codes. The underlying cause of death refers to that one which initiated the chain of pathological events that led to death, while the final cause of death is the disease or event that produced death directly (Department of Health and Human Services, 2003).

We primarily focused on non-communicable underlying causes of death: CKD, diabetes, COPD, HTN, and selected cancers (ICD-10 codes shown in the data sources and subjects section above); we also generated one category for all others causes of death. We focused on these non-communicable causes because they are leading causes of death (GBD 2016 Causes of Death Collaborators, 2017; Fitzmaurice et al., 2018; Perú Cancer Profile–PAHO, 2014). On the other hand, the final causes of death were classified in three groups: communicable, cardiovascular (CVDs) and neoplasms. Their ICD-10 codes were classified according to the Global Burden of Disease Study criteria (GBD 2016 Causes of Death Collaborators, 2017), though with the following modifications: communicable final causes also included unspecified pneumonia (ICD-10: J18.9), unspecified bronchopneumonia (ICD-10: J18.0), and unspecified sepsis (ICD-10: A41.9); CVDs also included stroke not specified as haemorrhage or infarction (ICD-10: I64.0), unspecified heart failure (ICD-10: I50.9), and congestive heart failure (ICD-10: I50.0). For a full description of ICD-10 codes included in the final causes, refer to Table S1.

We did not strictly follow the Global Burden of Disease classification because we would have experienced a massive loss of information; in fact, the codes that were included in addition to the Global Burden of Disease classification accounted for the largest proportion of reported deaths. We chose this approach based on our knowledge of the local context. For example, sepsis coded as unspecified sepsis (ICD-10: A41.9) would have been considered a “garbage code” by the Global Burden of Disease classification, but we included it in our communicable diseases category. We made this decision because we believe that these cases could have been undefined because of lack of laboratory resources to further characterize this condition. However, despite of this limitation, a sepsis would most likely be a communicable cause of death hence we included it in such group. For transparency sake, results according to the Global Burden of Disease classification are shown as Fig. S2.

Statistical analysis

Categorical variables are described as frequencies as well as proportions with 95% confidence intervals (95% CI), and comparisons between them were computed with the Chi2 test. Numerical variables were described as mean and standard deviation. A sensitivity analysis was conducted with the final causes of death as per the Global Burden of Disease Study criteria, which follow a more strict classification. Analyses were conducted with STATA 13.0 (StataCorp, College Station, TX, USA). Data was analyzed without further re-distribution of ill-defined ICD-10 codes.

Ethical considerations

De-identified data was used; hence privacy of individuals was protected. Because this work was of low risk to human subjects and data were freely available, approval from an Institutional Review Board was not sought.

Results

General characteristics

There were 77,065 studied observations, with a mean age at death of 69.4 (SD: 19.3) years ranging from 18 to 115; there were more men (53.9%). Of the selected underlying cause of death, diabetes accounted for 1.1% (95% CI [1.0–1.2%]), CKD for 2.0% (95% CI [1.9–2.1%]), COPD for 1.7% (95% CI [1.6–1.8%]), HTN for 2.4% (95% CI [2.3–2.5%]), and cancers for 6.4% (95% CI [6.2–6.6%]). There was a sex difference (p < 0.001), with more women deaths in the diabetes (1.2% vs 1.0%), COPD (1.8% vs 1.7%), HTN (2.5% vs 2.2%), and cancer (7.3% vs 5.7%) groups (Table S2).

Final causes of death

Communicable diseases accounted for the largest share (60.0%; 95% CI [59.4–60.5%]), followed by cardiovascular (29.8%; 95% CI [29.2–30.3%]) and cancers (10.3%; 95% CI [9.9–10.6%]). There was a sex difference (p < 0.001) with more women in the communicable (60.4% vs 59.5%) and cancer (11.2% vs 9.4%) categories (Table S3).

Figure 1 shows that when HTN was the underlying cause of death, the largest final cause was CVDs (82.3%). The other underlying causes of death had a non-trivial share of communicable diseases as final cause of death, with COPD showing the largest (86.4%) share, followed by CKD (79.3%), cancer (76.5%), and diabetes (69.3%, Table S4). Table 1 shows the top three final causes of death in each underlying cause.

Figure 1 Final cause of death in selected underlying causes, 2015.

This figure is to be read horizontally: starting from all deaths (first node or column), how many of these were due to the selected underlying causes of death (second set of nodes or second column), and of these how many were attributed to the studied final causes of death (last set of nodes or last column). The width of the links between nodes is relative to the proportion of deaths, that is, the wider the link the larger the proportion. Interpretation: from all deaths that occurred in 2015, the largest proportion was due to other causes (wider link going from all deaths to other causes); also, when hypertension was the underlying cause of death, the largest final cause fall in the cardiovascular category (wider link going from HTN to CVDs). CKD, chronic kidney disease; COPD, chronic obstructive pulmonary disease; HTN, hypertension; CVD, cardiovascular diseases. Refer to Table S5 for the exact estimates.

Table 1 Top three final causes of death according to each underlying cause.

Diabetes (N = 363)	
Communicable (N = 248)	Cardiovascular (N = 112)	Cancer (N = 3)	
Unspecified sepsis (28.3%)	Unspecified acute myocardial infarction (54.5%)	Malignant neoplasm of gallbladder (33.3%)	
Unspecified pneumonia (11.3%)	Heart failure (7.1%)	Malignant neoplasm of kidney, except renal pelvis (33.3%)	
Unspecified bronchopneumonia (3.2%)	Unspecified heart failure (7.1%)	Unspecified major salivary gland (33.3%)	
Chronic kidney disease (N = 511)	
Communicable (N = 405)	Cardiovascular (N = 103)	Cancer (N = 3)	
Unspecified sepsis (84.0%)	Unspecified acute myocardial infarction (39.8%)	Multiple myeloma (33.3%)	
Unspecified pneumonia (11.1%)	Heart failure (16.5%)	Oesophagus (33.3%)	
Unspecified bronchopneumonia (2.2%)	Unspecified heart failure (14.6%)	Unspecified myelodysplastic syndrome (33.3%)	
Chronic obstructive pulmonary disease (N = 382)	
Communicable (N = 330)	Cardiovascular (N = 52)	Cancer (N = 0)	
Unspecified sepsis (54.6%)	Unspecified acute myocardial infarction (48.1%)		
Unspecified pneumonia (32.4%)	Unspecified heart failure (17.3%)		
Unspecified bronchopneumonia (9.4%)	Heart failure (9.6%)		
Hypertension (N = 997)	
Communicable (N = 175)	Cardiovascular (N = 820)	Cancer (N = 2)	
Unspecified sepsis (72.6%)	Unspecified acute myocardial infarction (51.8%)	Unspecified colon (50.0%)	
Unspecified pneumonia (17.1%)	Stroke, not specified as hemorrhage or infarction (19.8%)	Unspecified myelodysplastic syndrome (50.0%)	
Unspecified bronchopneumonia (6.3%)	Heart failure (5.4%)		
Cancer (N = 1,189)	
Communicable (N = 910)	Cardiovascular (N = 147)	Cancer (N = 132)	
Unspecified sepsis (71.8%)	Unspecified acute myocardial infarction (49.7%)	Carcinoma in situ, stomach (20.5%)	
Unspecified pneumonia (15.1%)	Unspecified heart failure (20.4%)	Carcinoma in situ, prostate (14.4%)	
Unspecified bronchopneumonia (7.4%)	Heart failure (6.8%)	Carcinoma in situ, bronchus, and lung (6.8%)	
Note:

The Ns mean that, within each underlying cause of death, N people finally died of a communicable, cardiovascular or cancer cause. For example, refer to diabetes as underlying cause of death: among those whose underlying cause of death was diabetes, 248 finally died of a communicable cause, 112 of a cardiovascular cause, and three of a cancer; within diabetes as the underlying cause of death, and among those 248 who finally died of a communicable disease, 28.3% (of 248) died of unspecified sepsis, 11.3% of unspecified pneumonia, and 3.2% of unspecified bronchopneumonia (these were the top three final communicable causes of death where diabetes was the underlying cause). The same interpretation follows for the other underlying and final causes of death.

Sensitivity analysis

Figure S2 shows that in sensitivity analysis the pattern of final cause of death changed: all underlying causes had a large share of CVDs as final cause of death. The profile of underlying and communicable final cause of death remained similar; where COPD had the largest share (25.5%) of final communicable diseases, followed by cancer (18.5%), CKD (14.3%), and diabetes (8.0%, Table S4).

Discussion

Main findings

The results showed a divergent profile between the underlying and final cause of death. In four non-communicable underlying causes of death, namely CKD, COPD, diabetes, and cancer, the final cause of death was largely a communicable condition; however, when HTN was the underlying cause, cardiovascular events were the predominant final cause. These results suggest that, even though Peru is undergoing epidemiological transition with non-communicable diseases leading the mortality rankings (GBD 2016 Causes of Death Collaborators, 2017; Huicho et al., 2009), communicable diseases still play a non-trivial role in the final cause of death. This calls for a synergism between health policies and practice of non-communicable and communicable diseases. In addition, our findings could also hide a biological fact: reduced immunity in patients with underlying non-communicable diseases could ultimately lead to communicable illnesses. Improving nutrition among these patients, advising and providing them with information to avoid communicable diseases may be relevant, and even to strengthen vaccination coverage (e.g., influenza) for these patients as is the case with the elderly.

Accounting for its limitations, this work adds to the literature evidence from the general population, that is, not based on cohorts of specific diseases. This preliminary work based on death registries builds the first step to better characterize, hence understand, the complex interactions between non-communicable and communicable causes of death, that is, multi-morbidity (multi-mortality). The results do not suggest that Peru is at a different stage of the epidemiological transition. However, the results invite us to think beyond these global metrics and understand that, even though Peru is at a stage where non-communicable diseases are more prevalent, communicable disease still accounts for a non-negligible proportion of final causes of death. We therefore recommend focusing on strengthening care for non-communicable diseases, while also keeping efforts up against communicable illnesses.

Comparison with available evidence and interpretation of results

A population-based cohort in Mexico showed that the rate ratio comparing diabetes vs no diabetes was larger when the outcome was infections (respiratory or gastrointestinal) than when it was cardiovascular mortality (Alegre-Diaz et al., 2016). This observation agrees with our overall results because the final cause of death when the underlying cause was diabetes was largely communicable illnesses. This similarity probably reflects poor awareness, treatment and control in diabetic patients (Azañedo et al., 2017; Herrington et al., 2018; Huayanay-Espinoza et al., 2016; Lerner et al., 2013).

In Canada, a registry-based study found that in people with CKD the leading cause of death was cardiovascular events; in addition, they also reported that with worse kidney function, the risk of dying of an infectious cause increased (Thompson et al., 2015). This observation had also been retrieved in the USA and Canada studying pneumonia fatalities (James et al., 2009; Wang et al., 2011). To the best of our knowledge this evidence is lacking in Peru where CKD mortality has only been studied in hospitals and patients in renal replacement therapy (Herrera-Añazco, Pacheco-Mendoza & Taype-Rondan, 2016). A hospital-based study in northern Peru reported that the first cause of death in CKD patients was CVD followed by infections (Concepción-Zavaleta et al., 2015). It seems that our primary results disagree with the available evidence. This could be explained by ascertainment of the outcome variable. Studies in high-income countries and those in restricted hospital-based populations could have followed a stronger methodology to assess the outcome. This hypothesis is supported by our sensitivity analysis in which cardiovascular events seemed to be the largest final cause of death when CKD was the underlying cause. This discrepancy and potential explanation call for cautious interpretation of these findings and for strengthening the mortality reporting systems in Peru (Zolezzi, 2017).

A Spanish COPD patient-based cohort reported different causes of death according to COPD profile; of note, in none of the studied profiles did a communicable disease prevail over cardiovascular causes of death (Golpe et al., 2018). This disagrees with our primary results, which could be explained by outcome ascertainment methods as previously discussed. However, this could also hide other clinical patterns such as therapy. For example, COPD patients on inhaled corticosteroids may have worse outcomes, including death, in the presence of community-acquired pneumonia (Liu, Han & Liu, 2018). To the best of our knowledge there is no evidence on mortality or treatment patterns in COPD patients in Peru. Although largely speculative, if inhaled corticosteroids were the leading treatment, this could explain our results.

In the USA a registry-based study showed that breast and prostate cancer patients had a higher frequency of a non-cancer death (Zaorsky et al., 2017). In addition, the top non-cancer cause of death appeared to be cardiovascular conditions, the second cause was infectious illnesses (Zaorsky et al., 2017); moreover, they also found that in people younger than 45 years, infections accounted for a relevant share of deaths, whereas in older people, heart diseases surpassed infections (Zaorsky et al., 2017). Our primary results showed that communicable conditions were the leading cause of death, whereas our sensitivity analysis agreed with Zaorsky et al. (2017). The discrepancy could be due to different methods of assessing the outcome or different cardiovascular profiles between populations. With regards to the age finding, post-hoc analysis (data not shown in main results) yielded similar observations: for our primary outcome when the underlying cause of death was cancer and age was restricted to younger than 45 years, 80.5% eventually died of a communicable cause, whilst this percentage for people 45 years and above was 76.2%; on the other hand, when this post-hoc analysis was conducted with the outcome for sensitivity analysis, these percentages were 29.4% and 17.8%, respectively. These results also suggests that when there is an underlying cancer, infectious causes of death may be more frequent in younger than older people. This finding deserves further examination with larger sample sizes in both age groups to understand the profiles and risk factors of these sub-populations.

Hypertension is known to be a key risk factor for cardiovascular events, and our results confirm this knowledge. In addition, our results point that in some deaths with underlying HTN the final cause was a communicable disease. This could signal a complication of cardiovascular events, such as post-stroke infections (Castaneda-Guarderas et al., 2011; Westendorp et al., 2011).

Public health and clinical relevance

Our results show a divergence between the underlying and final cause of death, that is, non-communicable causes and eventually dying of a communicable illness. From a public health perspective these results could signal that CKD, COPD, diabetes, and cancer patients should be aware of how to control their underlying condition, but also of the potential communicable complications so that they can identify them early. In addition, the results may call for providing health personnel and facilities with resources to successfully address both communicable and non-communicable diseases in addition to these underlying conditions.

Strengths and limitations

The strengths of this work include using a large national death registry based on death certificates, and the analysis of both underlying and final causes of death. Nevertheless, the limitations included the lack of information on other health and socio-demographic variables such as adherence to treatment, disease control or educational attainment, which could modify and explain the patterns herein explored (Herrington et al., 2018). However, our results still provide evidence of a divergence between the underlying and final cause of death. Future studies should explore the determinants at the individual and public health level of this divergence. Second, information was not available about the events between the underlying and final cause of death, should there have been any. For example, for deaths that occurred in a hospital, the final cause could have been a communicable disease even though the hospitalization was due to a non-communicable illness (e.g., COPD patient hospitalized due to a coronary syndrome with acquired pneumonia as the final cause of death). Nonetheless, our results raise awareness that some non-communicable diseases may have a non-negligible chance of dying of a communicable disease, signaling that we should not lose sight of these diagnoses should not be lost even though other (non-communicable) conditions seemed more important at the time. This could be relevant in resource-limited settings where lack of human and logistical resources make it difficult to consider all potential possibilities.

Conclusions

In a middle-income country undergoing epidemiological transition, there seems to be a divergence between the overall profile of the underlying and final cause of death. In four major non-communicable diseases—CKD, COPD, diabetes, and cancers, the final cause of death was largely a communicable illness; whereas in HTN it was a cardiovascular event. Although efforts should be deployed to prevent underlying non-communicable diseases, potential communicable complications should not be neglected.

Supplemental Information

Supplemental Information 1 Supplementary Material.

Click here for additional data file.

Supplemental Information 2 Raw data.

Click here for additional data file.

Additional Information and Declarations

Competing Interests

Author Contributions

Data Availability

The authors declare that they have no competing interests.

Rodrigo M. Carrillo-Larco conceived and designed the experiments, analyzed the data, prepared figures and/or tables, authored or reviewed drafts of the paper, approved the final draft.

Antonio Bernabe-Ortiz conceived and designed the experiments, analyzed the data, authored or reviewed drafts of the paper, approved the final draft.

The following information was supplied regarding data availability:

The Raw data are available in the Supplemental Files.

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
