# Peer review of "A divergence between underlying and final causes of death in selected conditions: an analysis of death registries in Peru"

_PeerJ, doi:10.7717/peerj.5948_

## Round 0.1 · original submission · Major Revisions

As you will see the reviewers have very significant concerns with multiple aspects of this paper. They provide a very detailed explanation of their concerns so I do not need to prove additional guidance. Please read their comments and respond as best you can. Final acceptance is not assured.

·

Basic reporting

Some English language editing necessary throughout. For example in abstract: “the 53.9% was men.” is not grammatical. Likewise in line 90: “Two were the variables of interest”, and line 172 “evidence lacks in Peru”, etc.

Line 208: Should be "sensitivity" analysis.

Experimental design

The study is observational, so there is no experimental design. Nonetheless, representativeness and data quality are issues that deserve some discussion.

There is no mention of coverage of the death certificates for the population of Peru. 77,000 deaths in a country of 30 million people would be a death rate of a quarter of a percent in a year. The correct death rate is approximately 6 per 1000 per year, making the data in this study equal to about half of expected deaths. Where are the 50% of the deaths?
Also no mention of the quality of the death certificates, number of certificates with incomplete or impossible fields, etc.

Validity of the findings

The authors excluded many deaths if the final cause of death was the same as the underlying cause of death, which is worrisome. For example, breast cancer could not be both an underlying and acute cause of death. This is not justified by the authors, nor explained in any way. They promise a flowchart showing the number of these exclusions in a supplementary figure, but I could not find this figure.

Line 95: Final causes of death were defined as “communicable, cardiovascular (CVDs) and neoplasms”. I don’t understand what happens to external causes of death (e.g. accident, homicide, suicide, overdose, etc).

Additional comments

The authors note the importance of mortality statistics in international comparisons, as a broad indicator of health, social policy and medical systems. They study the underlying-final cause of death pattern in one LMIC (Peru).

The abstract states “The underlying-final cause of death pattern has not been explored at the country level” but this statement is not true in general, only true of Peru. In rich countries this is not an unusual analysis. Even globally, while it seems that no paper has specifically targeted Peru, many papers using the WHO mortality database do include Peru data (e.g. PMID 29717336), and many papers focus on other LMICs in the region (e.g. PMID 22124693) or in specific cities in the region (e.g. PMID 21986787).

Background: The authors discuss underlying versus acute causes of death as registered on death certificates, but they do not cite a large literature on this topic, for example, discussing the methods by which underlying causes are inferred (PMID 27127419, 29060620, 27994280) or the historical evolution of these conventions and their intended interpretations (e.g. Jentzen JM Death Investigation in America. Harvard, 2009).

Discussion: The authors conclude that there is a divergent profile between underlying and final causes of death. It is implied that this is somehow unusual, but the is little justification provided to suggest that this is a novel or informative observation. Is the same pattern common in rich countries, or in other LMICs in which it has been examined? The authors do mention a few specific examples, such as a study of diabetes in Mexico, COPD in Spain, and kidney disease in Canada, but no general principles regarding the relations between underlying and acute causes.

For example, that people with chronic illnesses often die of pneumonia, sepsis, hospital acquired infections or other communicable cause is hardly surprising. It does not seem to relate to the epidemiologic transition, as suggested by the authors. Rather it seems to be a natural consequence of reduced immunity, exposure to hospitals, bedsores, injections, etc.

Public health and clinical relevance: The authors note a divergence between the underlying and final cause of death as the primary insight, but at least some of this was iatrogenic, since they purposefully deleted many observations if these fields matched.

The authors seem to find that the significance of this analysis is to reinforce the concept that a non-communicable disease is not the only possible complication over an underlying non-communicable condition. I can’t imagine that any clinician ever thought otherwise. For example, about a quarter of all Alzheimer’s patients die acutely of a respiratory infection (PMID: 27192207). This pattern is therefore of no particular surprise.

Reviewer 2 ·

Basic reporting

Basic Reporting
1. The main argument for the importance of this study is the differentiation of final vs underlying causes of death. The authors make three points in the introduction, related to the public health, clinical and epidemiological transition relevance of this differentiation. While the clinical and epidemiological transition ones are well explained, I fail to understand the public health argument: “For example, whether a patient with type 2 diabetes is likely to finally perish of a coronary event or sepsis, could guide what information should be provided to these patients. Information could thus target prevention of non-communicable diseases, communicable disease, or both”. How is this a public health argument, when it relates to patients and information? If this is a clinical argument, then this paragraph should be reworded to avoid this confusion.
2. Moreover, and related to comment #1 above, the authors need to define how “final” and “underlying” causes of death are defined in death certificates and what are the instructions for professionals filling them.
3. The English language should be improved to ensure that an international audience can clearly understand the text. A few examples from the abstract: the second sentence is missing a word (or ultimately should be replaced with other word); the second sentence of results is odd (“the 53.9% was men”); the last sentence of the results section gives the impression that COPD, CKD, cancer and diabetes are all communicable diseases.
4. The sensitivity analysis lacks detail. How are the GBD categorizations different? It is hard to interpret the meaning of this analysis without knowing the difference between the author’s and GBD’s categorization.
5. The Sankey diagram of Figure 1 is quite hard to read and interpret. “All deaths” is smaller than any of its subcomponents, which are then similar in size to the third level. The authors detail that they have used proportions, but the figure is deceptive by not using a height that is proportional to the number of deaths across columns. It is also unclear (just from looking at the figure) that the middle ones are final causes of death and the right ones are underlying causes, so an horizontal axis would help (although this is included in the footnote).
6. Table 1: what is the meaning of the N=<number> rows? Are these the number of unique final causes of death? In that case, this should be made explicit. Moreover, table 1 could contain more information to be more useful. For example, by looking at table 1 it’s hard to see the contribution of each final cause of death to the OVERALL amount of deaths in each underlying cause. See for example Cancer (final?) and diabetes (underlying?). There are 3 unique final causes of death, but it’s hard to see in the table whether cancer makes up a sizable amount of the deaths with diabetes as an underlying cause. It’d be more informative if the N=<number of unique causes?> was replaced with the proportion of deaths in each large group of final causes. Even better, it’d be good if the authors showed raw numbers and overall proportions, so that the proportion of “Malignant neoplasm of gallbladder” is actually the % of deaths with diabetes as underlying with that cancer as the final cause of death.
7. The first reference (Fitzmaurice 2018) refers only to cancer burden. I suggest the authors reference the general GBD papers instead of the cancer specific ones.

Experimental design

Experimental Design
8. The authors mention underreporting briefly, but according to the Peruvian ministry of health (http://bvs.minsa.gob.pe/local/minsa/2722.pdf), PAHO (http://iris.paho.org/xmlui/bitstream/handle/123456789/34329/CoreIndicators2017_eng.pdf?sequence=1&isAllowed=y), and others (https://www.ssc.wisc.edu/cdha/latinmortality/) this happens in around 40-43% of the deaths, with some Departamentos (such as Loreto) having around 80% of under-reported deaths. The authors need to devote some time to explaining how this underreporting may affect their results. For example, if this underreporting is differential by age or sex, then some final causes of death may be more underreported than others. Or, if areas with more underreporting have a higher degree of communicable diseases, then results may be biased.
9. The authors do not justify their selection of deaths in people aged 18 or above. My guess is that this is because they have chosen deaths uncommon in children, but this should be made explicit.
10. The authors exclude cardiac arrest deaths (coded as a final cause), which are coded as garbage codes by GBD. The authors do not also detail what was done with other garbage codes as defined by GBD (such as R codes). Are these just excluded? Some of them have been included (like “unspecified sepsis” or “unspecified heart failure”), but it is unclear what criteria was used for this.
11. It is unclear why authors exclude deaths with missing location of death (see Supplementary Figure 1). Moreover, does this refer to the district (geographical location) or type of institution (like hospital vs home)?.

Validity of the findings

Validity of Findings
12. My main concern with the validity of the findings is that the authors stress that this shows that Peru is in a different stage of the epidemiologic transition, but offer no evidence to compare Peru with other countries in this part of the manuscript. How is this “divergence” different in other countries? Are the distributions of final causes of death different by underlying cause in other countries? In fact, the authors compare the specific results of diabetes to Mexico and show that their results are similar, and some of the results in USA/Canada/Spain are similar too. I think the authors need to do a more systematic comparison here
13. My second comment on validity is related to the first: isn’t this divergence between final and underlying causes of death occurring by design? Since the authors do not devote much time to the conceptual difference between them it’s hard to see if this is an expected outcome.
14. My third comment on validity is related to comment (2) in basic reporting above: are instructions to fill final/underlying causes of death the same in Peru as compared to other countries? If instructions are different, then these results may be the result of measurement differences and not differneces in underlying public health. An internal comparison in Peru over time may be more interesting in this case.

Additional comments

PeerJ-30352
Title: A divergence between underlying and final causes of death in selected conditions: An analysis of death registries in Peru

This study leverages the reporting of final and underlying causes of death in mortality registries in Peru, in order to understand what final causes of death were most common among 9 selected underlying causes of death. The study seems interesting for people interested in understanding the burden of disease using death records. However, I believe the authors need to improve the reporting of the study and work on changing unsubstantiated claims that are not related to the study findings. Comments are structured in basic reporting, experimental design, and validity of findings.

---

## Round 0.2 · accepted · Accept

The improvements, while not curative, are sufficient.

·

Basic reporting

The English language in the revised submission is adequate. Basic reporting issues seem sound.

Experimental design

no experimental design (not applicable)

Validity of the findings

I still question the utility of the findings given the underreporting of deaths, and there is no quantitative attempt in the paper to confront the potential bias if these are differentially unreported, as seems likely (by region, by cause, by social factors, etc). But the authors have at least been transparent about this concern.

Additional comments

Building capacity for demographic surveillance in Peru is an important goal. Building reporting systems will probably be necessary before real improvements in data quality can be realized, but this at least is a first step.